# Colorimetric Analysis of Glucose Oxidase-Magnetic Cellulose Nanocrystals (CNCs) for Glucose Detection

**DOI:** 10.3390/s19112511

**Published:** 2019-05-31

**Authors:** Ying Chuin Yee, Rokiah Hashim, Ahmad Ramli Mohd Yahya, Yazmin Bustami

**Affiliations:** 1School of Biological Sciences, Universiti Sains Malaysia, Penang 11700, Malaysia; yychuin@gmail.com (Y.C.Y.); armyahya@usm.my (A.R.M.Y.); 2Division of Bioresource, Paper and Coatings Technology, School of Industrial Technology, Universiti Sains Malaysia, Penang 11700, Malaysia; hrokiah@usm.my

**Keywords:** cellulose nanocrystals, iron oxide nanoparticles, magnetic cellulose nanocrystals, immobilization, glucose oxidase, detection

## Abstract

Glucose oxidase (EC 1.1.3.4) sensors that have been developed and widely used for glucose monitoring have generally relied on electrochemical principle. In this study, the potential use of colorimetric method for glucose detection utilizing glucose oxidase-magnetic cellulose nanocrystals (CNCs) is explored. Magnetic cellulose nanocrystals (magnetic CNCs) were fabricated using iron oxide nanoparticles (IONPs) and cellulose nanocrystals (CNCs) via electrostatic self-assembly technique. Glucose oxidase was successfully immobilized on magnetic CNCs using carbodiimide-coupling reaction. About 33% of GOx was successfully attached on magnetic CNCs, and the affinity of GOx-magnetic CNCs to glucose molecules was slightly higher than free enzymes. Furthermore, immobilization does not affect the specificity of GOx-magnetic CNCs towards glucose and can detect glucose from 0.25 mM to 2.5 mM. Apart from that, GOx-magnetic CNCs stored at 4 °C for 4 weeks retained 70% of its initial activity and can be recycled for at least ten consecutive cycles.

## 1. Introduction

Glucose oxidase sensors have been used extensively for glucose level monitoring. Generally, the commercialized glucose sensors operate under electrochemical principle, in which an oxygen electrode is required to detect the amount of oxygen consumed for the reaction of glucose oxidase and glucose molecules [1]. Recently, colorimetric method has been a favorable approach due to several factors such as low cost, convenience, and ease-of-use [2]. Since the discovery of nanoparticles, most of the colorimetric approaches utilize plasmonic sensing platforms, such as silver nanoparticles (AgNPs), gold nanoparticles (AuNPs), and quantum dots to detect the presence of glucose based on the shift of localized surface plasmon resonance (LSPR) wavelength due to the liberation of hydrogen peroxide (H_2_O_2_) upon reaction of glucose oxidase with glucose molecules. However, LSPR technique suffers from background interference [2]. Therefore, instead of utilizing LSPR shift for detection, a colorimetric sensing strategy based on the oxidation of chromogenic substances is investigated in this study.

Glucose oxidase is an enzyme that functions to catalyze the oxidation of glucose without being used up in the reaction. Therefore, it is useful to recover and reuse. To achieve this, glucose oxidase can be immobilized on a support. As the surface area of the support largely affects enzyme loading, magnetic cellulose nanocrystals (magnetic CNCs) with large surface area are seen with great interest as an immobilization platform. Several immobilization techniques have been established, and generally can be categorized as physical and chemical methods. Physical methods involve weak interaction between support and enzyme while chemical methods involve strong covalent bonding between support and enzyme [3]. A physical method such as adsorption is a common non-covalent approach to immobilize enzymes but suffers from enzyme leakage when there are slight changes in the reaction environment, such as temperature and pH [4]. Due to this reason, covalent bonding is preferable in order to prevent enzymes detaching from its support.

The magnetic CNCs used in this study are a combination of cellulose nanocrystals (CNCs) and iron oxide nanoparticles (IONPs). CNCs is a nanomaterial isolated from various cellulosic resources, such as cotton, lignocellulose, and marine creatures such as tunicates, algae, and bacteria [5]. One of the main strategies for CNCs isolation is using acid hydrolysis, specifically sulfuric acid since it produces readily dispersed CNCs [6,7]. The property such as large surface area, high specific strength, and low toxicity make as an attractive candidate in the field of biomedicine, biosensors, and biocatalysis [5,8,9,10]. In addition, the abundance of hydroxyl (–OH) group on the surface of CNCs allows for simple and easy surface modification, serving as an ideal attachment platform in various applications, including enzyme immobilization. In recent years, several enzymes such as lipase [9], glucose oxidase [11], peroxidase [12], and papain [13] have been successfully immobilized on CNCs. Furthermore, incorporating IONPs with CNCs can facilitate an easy separation of immobilized enzyme from the reaction mixture, thus permitting reusability.

There are few techniques that have been utilized for fabrication of magnetic CNCs such as self-assembly, in situ synthesis method, and carbodiimide coupling method [13,14,15,16,17]. The most favorable method for preparing magnetic CNCs is through self-assembly technique that involves the electrostatic interaction between magnetic nanoparticles and CNCs. Since both CNCs and IONPs are negatively-charged, the study in Reference [14] modified CNCs surface with chitosan, a positively charged polysaccharide, so that electrostatic interaction was established between positively charged CNCs-chitosan and negatively charged Fe_3_O_4_, thus forming magnetic CNCs. Instead of modifying CNCs, Reference [13] successfully modified the Fe_3_O_4_ surface using a positively charged polymer, polyetyleneimine (PEI), prior to self-assembly with CNCs. Interestingly, both studies showed the potential use of magnetic CNC for enzyme immobilization, in which an enzyme could be easily separated and recycled from the reaction mixture through external magnetic forces.

In our study, magnetic CNCs were first fabricated and characterized, then used as a platform for glucose oxidase immobilization and detection of glucose. The magnetic CNCs were fabricated using electrostatic self-assembly technique without the addition of polymers or polysaccharides on the surface of nanomaterials. Instead, the IONPs’ amphoteric nature were manipulated, as IONPs’ surface charges can be easily tuned to positive or negative values by a simple pH adjustment and, allowing direct attachment with negatively-charged CNCs. The characteristics of magnetic CNCs were analyzed using TEM, zeta potential, and FTIR. Immobilization of glucose oxidase was performed via carbodiimide-coupling technique to establish a covalent bond between glucose oxidase and magnetic CNCs. Subsequently, the potential of glucose oxidase-magnetic CNCs for glucose detection were analyzed based on ABTS colorimetric analysis.

## 2. Materials and Methods

### 2.1. Preparation of Magnetic CNCs

Magnetic CNCs were prepared by electrostatic self-assembly between CNCs and IONPs. CNCs were prepared from vascular bundle of oil palm (*Elaeis guineensis*) trunk using sulfuric acid hydrolysis process as described in Reference [18]. This was followed by 2,2,6,6-tetramethylpiperidine-1-oxyl (TEMPO)-mediated oxidation according to the method developed by Reference [19] with slight modification to change the surface –OH group into –COOH group. Approximately 1 g of CNCs were dispersed in distilled water using homogenizer (IKA T10 basic, Sigma). Then, 0.0125 g of TEMPO and 0.125 g of sodium bromide (NaBr) were added to the CNCs suspension and were mixed at room temperature. To initiate the oxidation process, approximately 9 mL of sodium hypochlorite (NaClO) was added dropwise to the CNCs suspension. 0.5 M sodium hydroxide (NaOH) was added to the suspension to maintain the pH around 10 to 11 throughout the reaction. Once pH of the suspension was stable, about 10 mL of ethanol was added to stop the reaction. Finally, CNCs were washed thoroughly with distilled water (dH_2_O) and freeze-dried. On the other hand, IONPs were synthesized by reverse co-precipitation technique as reported in Reference [20]. After that, approximately 50 mg of CNCs were dispersed in 50 mL of deionized water and 10 mL of IONPs were added and stirred vigorously at room temperature for 2 h. The resulting hybrid nanomaterials were then separated with an external magnet and re-dispersed in 50 mL of deionized water for further use.

### 2.2. Characterization of as-Prepared Nanomaterials

The size and morphology of CNCs, IONPs, and magnetic CNCs were observed using transmission electron microscopy (TEM) (Philips CM12). Each TEM sample was typically prepared by dropping sample suspension on a 400 mesh Cu grid with diameter of 3.05 nm. CNCs and magnetic CNCs were stained with 2% phosphotungstic acid. Fourier transform infrared (FTIR) was carried out with Shimadzu IRPrestige-21 in transmittance mode using the potassium bromide (KBr) pellet technique. Zeta potential was estimated using Malvern Zetasizer Nano Instrument ZS, in which each sample were prepared at concentration around 1 mg/mL.

### 2.3. Immobilization of Glucose Oxidase on Magnetic CNCs

Prior to glucose oxidase (GOx) immobilization, the –COOH group of magnetic CNCs were activated using carbodiimide-coupling reaction (EDC/NHS). About 1 mg of magnetic CNCs was resuspended in 1 mL 2-(*N*-morpholino)ethanesulfonic acid (MES) buffer (10 mM, pH 4.0). In a separate tube, 0.35 µL of 1-(3-dimethylaminopropyl)-3-ethylcarbodiimide (EDC) and 1.24 mg of *N*-hydroxysuccinimide (NHS) were mixed in 1 mL MES buffer (10 mM). After that, this solution was added to magnetic CNCs and was incubated at room temperature for 15 min. After incubation, magnetic CNCs were washed with PBS twice to remove excess reactants, then stored in buffer solution for GOx immobilization. For GOx immobilization, GOx from *Aspergillus niger* (EC 1.1.3.4) was purchased from Sigma-Aldrich, Inc and reconstituted in sodium acetate buffer (50 mM, pH 5). About 180 µL of activated magnetic CNCs was mixed with 20 µL of GOx (60 µg/mL) and incubated for 1 h at 4 °C. Following this, GOx-attached magnetic CNCs were recovered, washed with PBS twice, and stored in acetate buffer (10 mM, pH 5.0) for further use. The presence of GOx was confirmed by sodium dodecyl sulfate-polyacrylamide gel electrophoresis (SDS-PAGE). The supernatant (containing unbound GOx) were used to estimate the amount of GOx attached using Bradford assay.

#### 2.3.1. SDS-PAGE

SDS-PAGE was performed using 10% separating gel and 4% stacking gel. This protocol was adapted from the Bio-Rad User Manual. The gel buffer was prepared using 10% separating gel (2.5 mL separating buffer (1.5 M Tris-HCl, pH 8.8), 2.5 mL bis-acrylamides (40%), 100 µL ammonium persulfate (APS) (10%), 10 µL tetramethylethylenediamine (TEMED) and 4.89 mL dH_2_O) and 4% stacking gel (1 mL stacking buffer (1.0 M Tris-HCl, pH 6.8), 0.5 mL bis-acrylamides (40%), 50 µL APS (10%), 5 µL TEMED and 3.45 mL dH_2_O). For sample preparation, about 15 µL of each sample was aliquoted and mixed with 3 µL of loading dye and boiled for 5 min. About 15 µL of samples were loaded into each well and 5 µL of protein ladder (SMOBIO) was used to estimate the molecular mass of enzyme. Gels were run on a Bio-Rad Mini-PROTEAN Tetra cell gel electrophoresis unit. The gel was viewed under a calibrated imaging densitometer (GS-800, Bio-rad, Hercules, CA, USA).

#### 2.3.2. Bradford Assay

The amount of immobilized GOx on magnetic CNCs was determined using Bradford protein assay. The standard curve of known GOx concentration were prepared as follows: 160 µL of known GOx concentrations (20, 40, 60, 80, 100 µg/mL) were added to 40 µL of Bio-rad dye reagent concentrate in a 96-well microplate and was measured at 595 nm. For determination of immobilized GOx concentration, about 160 µL of the supernatant (containing unbound GOx) was added with 40 µL of dye reagent concentrate, assuming that all unbound GOx were present in the supernatant. After 5 min, the absorbance of dye solution was measured at 595 nm. The immobilization efficiency was calculated based on Equation (1):(1)$$\mathrm{Immobilization}\;\mathrm{efficiency}\;\left(\%\right)=100-\left(\frac{\mathrm{absorbance}\;\mathrm{of}\;\mathrm{unbound}\;\mathrm{GOx}}{\mathrm{absorbance}\;\mathrm{of}\;\mathrm{free}\;\mathrm{GOx}}\times 100\right)$$

#### 2.3.3. Determination of Immobilized GOx Efficiency

The reaction rate was determined by ABTS assay by varying the concentration of glucose from 5 mM to 100 mM. In a typical experiment, the assay was conducted by mixing 160 µL of acetate buffer (10 mM, pH 5.0), 10 µL of horseradish peroxidase (HRP) (12.5 µg/mL), 10 µL of ABTS (9.1 mM), 10 µL of glucose solution, and 10 µL of immobilized GOx. The reaction was carried out for 15 min at room temperature and the activity was calculated based on the absorbance measured at 415 nm at 1-min interval. The Michaelis–Menten constant (K_m_) and the maximum reaction rate (V_max_) were calculated by the following equation:(2)$$V=V_{\max}\frac{\left[S\right]}{\left[S\right]+K_{m}\;}$$
where V is the rate of oxidation reaction, V_max_ is the maximal rate of reaction, [S] is the concentration of glucose, and K_m_ is the Michaelis–Menten constant.

### 2.4. Glucose Detection Using ABTS Assay

Colorimetric detection of glucose underwent two separate reactions (3) and (4). Hydrogen peroxide (H_2_O_2_) produced upon reaction (3) served as the substrate for reaction (4) and produced oxidized ABTS, which generates a green coloration that has an absorption maximum at 415 nm. Glucose detection were carried out under conditions as described in Section 2.3.3. Limit of detection was calculated based on Equation (5).
(3)$$D-\mathrm{glucose}+H_{2}O+O_{2}\overset{\mathrm{GOx}}{\to }\mathrm{Gluconic}\;\mathrm{acid}+H_{2}O_{2}$$
(4)$$H_{2}O_{2}+{\mathrm{ABTS}}_{\left(\mathrm{red}\right)}\overset{\mathrm{HRP}}{\to }H_{2}O+{\mathrm{ABTS}}_{\left(\mathrm{ox}\right)}$$
(5)$$\mathrm{Limit}\;\mathrm{of}\;\mathrm{detection}=\frac{3.3\times \mathrm{standard}\;\mathrm{deviation}\;\mathrm{of}\;\mathrm{blank}}{\mathrm{slope}}$$

### 2.5. Specificity Test

For the specificity test, other types of sugar were used—sucrose (25 mM), maltose (25 mM), and fructose (25 mM), under conditions as described in Section 2.3.3.

### 2.6. Reusability and Storage Stability of Immobilized GOx for Glucose Detection

The enzymatic activities were studied under the condition as described in Section 2.4 using glucose (25 mM). The reusability of immobilized GOx were measured for ten consecutive cycles. After each run, the immobilized GOx was magnetically separated and washed with PBS twice to remove any remaining products. The residual enzyme activity after each cycle was measured and the relative activity was calculated based on the ratio of residual activity to the initial activity. For stability study, immobilized GOx were stored at 4 °C and the activity was measured every week for four weeks. For both experiments, the initial activity was assumed to be 100%.

## 3. Results and Discussion

### 3.1. Characterization

The morphology of CNCs, IONPs and magnetic CNCs were viewed using TEM. From Figure 1a, the prepared CNCs were rod-shaped-like particles with average length 273.9 ± 47.31 nm and width 8.5 ± 2.71 nm, respectively. This result is in agreement with the previous literature stating that the length of CNCs normally falls within the range of tens of nanometers to several micrometers, while width ranges from 3 nm to 50 nm [21]. As depicted in Figure 1b, IONPs were spherical in shape with diameter of approximately 12.9 ± 6.56 nm, slightly larger from the previous study [20]. The TEM image in Figure 1c shows that IONPs were successfully attached on CNCs since sphere grains of IONPs are clearly seen attached along the nanorods CNCs but the distribution was not uniform. Interestingly, the morphology of the magnetic CNCs remained unchanged as compared to their individual nanomaterials, CNCs and IONPs. Therefore, it was speculated that IONPs were attached randomly on the CNCs surface due to the ionic interaction scattered along the CNCs’ rod structure.

The zeta potential value of each studied nanomaterial is shown in Table 1. The zeta potential for CNCs is −42.53 mV whereas the zeta potential for acid-treated IONPs is +28.37 mV. Since both CNCs and IONPs were oppositely charged, a strong electrostatic attraction became the driving force for the formation of the hybrid nanomaterials, without the need of polymers or polyelectrolyte to mediate the electrostatic interaction. Furthermore, the results show that magnetic CNCs is negatively charged (−22.97 mV), in which it can be considered as metastable nanoparticles because the zeta potential value was lower than ±30 mV, which may contribute to slight aggregation within the solution. This value is about half the zeta potential value of CNCs, and it can be speculated that almost half of the CNCs charged sites were filled with the positively charged IONPs. The negative value is possibly contributed by the unbound carboxylic acid group present on the magnetic CNCs’ surface. These unbound carboxylic acid groups can be manipulated by attaching with other compounds, thus allowing multiple site attachment.

Figure 2 shows the analysis of the FTIR spectra of unmodified CNCs, modified CNCs and magnetic CNCs were analyzed across the regions of wavenumber 500 cm^−1^ to 4000 cm^−1^. Unmodified CNCs is denoted as CNCs before TEMPO-mediated oxidation. From this figure, five common peak regions were found. First, region (a) (3200 cm^−1^ to 3500 cm^−1^) arose from the stretching vibrations of hydroxyl (–OH) groups [13,18]. Second, peak (b) (2900 cm^−1^) was related to the symmetrical vibration of C–H bonds of the methyl (–CH_s_) groups present within the structure of CNCs [18]. Third, peak (c) (1373 cm^−1^) was due to C–H asymmetric deformation [18,22]. Fourth, region (d) (1030 cm^−1^ to 1200 cm^−1^) corresponded to C–O stretching region. C–O and C–O–C stretching vibrations are usually very strong and resulted in the formation of complex bands within this region [23]. Lastly, peak (e), observed at 898 cm^−1^, was due to the stretching vibration of β-glycosidic linkages between each cellulosic unit [18,24,25].

Besides, it was observed that all samples formed a peak around 1640 cm^−1^, which was associated with the O–H bending vibration of absorbed water during the isolation process of CNCs. It was found that after TEMPO-mediated oxidation was introduced, a more pronounced peak at 1616 cm^−1^ was observed, which may be attributed by the stretching vibration of C=O of sodium carboxylate (–COONa), indicating that the –OH groups on the CNCs’ surface were successfully displaced with –COONa groups. After the interaction of IONPs with CNCs, the peak at 1616 cm^−1^ shifted to 1716 cm^−1^, which indicated the presence of free carboxyl (–COOH) groups. In addition, the peak at 663 cm^−1^, representing C–OH bonds on CNCs, shifted slightly towards 667 cm^−1^ after TEMPO-mediated oxidation and shifted back to 663 cm^−1^ after fabrication with IONPs. This analysis suggested that the formation of magnetic CNCs possibly occurs on the –COONa groups on CNCs, where the transition of –COONa to –COOH groups were observed after IONPs were introduced. The presence of –COOH group will conveniently allow covalent attachment with GOx.

### 3.2. Immobilization of Glucose Oxidase on Magnetic CNCs

Immobilization of GOx was achieved through the formation of covalent bonds between GOx and magnetic CNCs. Covalent bonding is preferred, as it provides a stable immobilization system, since GOx will not be easily detached from the surface of magnetic CNCs as compared to physical method [11]. The formation of covalent attachment is due to the presence of carboxyl functional group (–COOH) on the magnetic CNCs surface and the amine group (–NH_2_) from enzyme GOx. The –COOH groups were initially activated through carbodiimide-coupling reaction using EDC/NHS that subsequently facilitated nucleophilic attack of –NH_2_ group of GOx. The successful immobilization of GOx on surface of magnetic CNCs was confirmed by SDS-PAGE analysis (Figure 3). It is clearly seen that no band was observed for activated magnetic CNCs (lane A), suggesting that the size of magnetic CNCs is too large to diffuse into the gel, hence, does not contribute to any bands. After GOx immobilization, a single band appeared in between 60–75 kDa (lane C) and in concomitant with the control band (free GOx) in lane B. This indicates that during sample preparation, GOx might be detached from magnetic CNCs and diffuse into the gel, resulting in the appearance of GOx band in lane C. GOx used in this study is a dimer of molecular weight 160 kDa. In this case, GOx migrates in its monomeric form, and the apparent molecular weight is slightly less than 80 kDa. The slight decrease in molecular weight might have been caused by the denaturation step prior loading into the wells. The result is found to be similar with Reference [26]. For quantification of the amount of immobilized GOx on the magnetic CNCs, Bradford assay was performed. By addition of 60 µg/mL of GOx to magnetic CNCs, only 20.08 µg/mL (estimated by Bradford assay) were successfully immobilized, thereby explaining the faded bands on Figure 3 after immobilization process.

To confirm the activity of GOx-magnetic CNCs, production of H_2_O_2_ during the oxidation of glucose was determined. Glucose oxidase catalyzes the reaction of glucose to gluconic acid and H_2_O_2_. H_2_O_2_ could oxidize ABTS in the presence of HRP to produce a green solution. Hence, H_2_O_2_ production can be measured based on the intensity of green coloration at 415 nm absorbance. The kinetic parameters (K_m_ and V_max_) of GOx before and after immobilization was determined from Lineweaver–Burk plot, as shown in Figure 4. The K_m_ value of immobilized GOx (6.38 mM) was one-third lower than that of free enzymes (9.03 mM). The decrease in K_m_ value after immobilization suggested that immobilized GOx has slightly higher affinity towards glucose molecules as compared to free enzymes, which may be due to the distribution and orientation of GOx on the surface of magnetic CNCs that provide greater accessibility of glucose molecules towards the active site of GOx. The improvement of affinity after enzyme immobilization was also observed when using α-amylase adsorbed on magnetic nanoparticles for starch hydrolysis [27] and cellulase immobilized on TiO_2_ nanoparticles for cellulose hydrolysis [28]. However, the V_max_ value was not further discussed as it was difficult to determine the V_max_ value accurately as the amount of immobilized GOx was estimated solely based on Bradford assay.

### 3.3. Glucose Detection

To evaluate the performance of GOx-magnetic CNCs for glucose detection, the absorbance of oxidized ABTS was monitored in the presence of different glucose concentration. Since IONPs was previously reported to possess intrinsic peroxidase-like activity [29], a control experiment in the absence of HRP was performed and no significant color change was observed (Figure 5). Therefore, the color change reaction in this study was solely dependent upon on HRP. With the increase of glucose concentration, the absorbance increases as well. The absorbance was proportional to glucose concentration in the range of 0.25 mM to 2.5 mM with a correlation coefficient of 0.9956 (Figure 5). The limit of detection of this detection system is 0.083 mM.

### 3.4. Specificity Test

To evaluate the effect of immobilization towads the specificity of glucose detection, fructose, sucrose, and maltose were used. Based on Figure 6, a significant amount of H_2_O_2_ production was detected when glucose was used as the substrate. Reference [30] shows the performance of free GOx, in which it rapidly oxidized glucose and requires a few hundreds times as much GOx to catalyze the oxidation of other type of sugars. Therefore, the high specificity of GOx-magnetic CNCs towards glucose as compared to other type of sugar suggested that the immobilization process did not affect the specificity of GOx.

### 3.5. Reusability and Storage Stability

The possibility of re-using the GOx-magnetic CNCs for ten successive glucose measurements was investigated. As seen in Figure 7a, GOx-magnetic CNCs maintained 100% of its initial activity even after ten reaction cycles. This may be attributed to the high dispersibility of magnetic CNCs in preventing agglomeration and also to the strong magnetic behavior of magnetic CNCs in minimizing loss of immobilized GOx during repeated magnetic separation and redispersion step. Besides that, reaction components mixed individually for each cycle led to a slight decrease in activity at cycle 3 and error bar extended beyond 100%. Table 2 compares the reusability of GOx magnetic CNCs in this study with other reported works. It is obviously shown that magnetic CNCs provides an excellent platform for enzyme immobilization since the immobilized GOx can be completely recovered and reused. These findings further support that covalent bonding between GOx and magnetic CNCs is significantly strong and can prevent enzymes detached from the platform, hence, maintaining its catalytic activity. Furthermore, by comparing to a study conducted by Reference [13], enzyme leakage was observed after four reaction cycles when immobilization of GOx was achieved through adsorption. To study the stability for long-term storage, GOx- magnetic CNCs were stored at 4 °C for four weeks. As shown in Figure 7b, after 1 week of storage, the activity drops significantly to 70% and remains unchanged throughout four weeks of incubation. Further investigation on the storage stability may be performed by addition of suitable additives in the storage buffer [31].

## 4. Conclusions

Magnetic CNCs were successfully fabricated based on electrostatic self-assembly between acid-treated IONPs and carboxylated CNCs, without the need of additional polymers to mediate the electrostatic interaction. The as-prepared magnetic CNCs with –COOH groups on its surface conveniently allows for covalent immobilization of GOx. Approximately 33% of added GOx was immobilized on magnetic CNCs and based on the Lineweaver–Burk plot, GOx-magnetic CNCs shows better affinity towards glucose molecules as compared to free GOx. In addition, the GOx-magnetic CNCs can specifically detect glucose within the range of 0.25 mM and 2.5 mM. Apart from that, GOx-magnetic CNCs demonstrated good stability for long-term storage and can be completely recovered and reused by its magnetic properties for at least ten consecutive cycles.

## Figures and Tables

**Figure 1 sensors-19-02511-f001:**
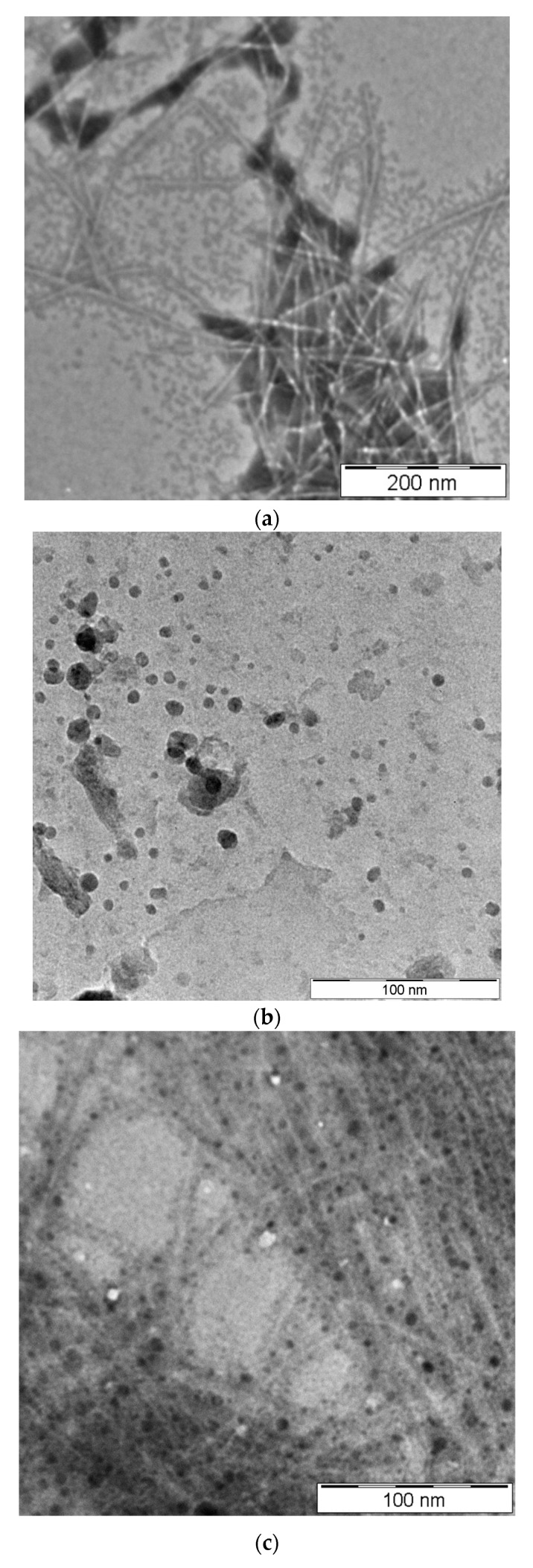
TEM images of (**a**) CNCs, (**b**) IONPs, and (**c**) magnetic CNCs.

**Figure 2 sensors-19-02511-f002:**
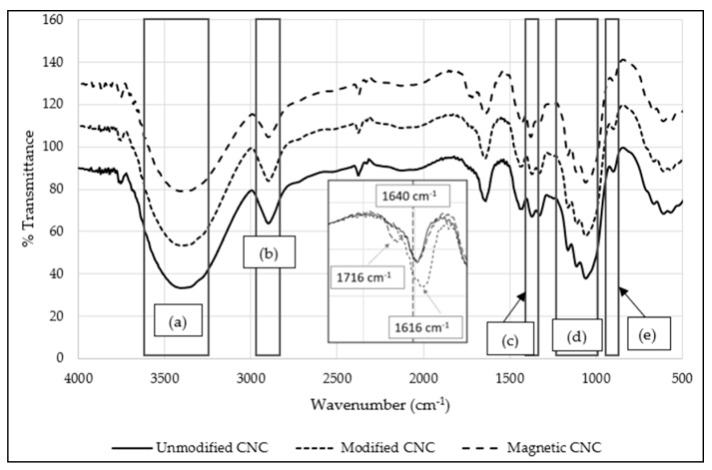
FTIR spectra of CNCs, modified CNCs and magnetic CNCs. (Inset showing expanded portion of wavelength 1600–1800 cm^−1^, where the peak shift is clearly seen).

**Figure 3 sensors-19-02511-f003:**
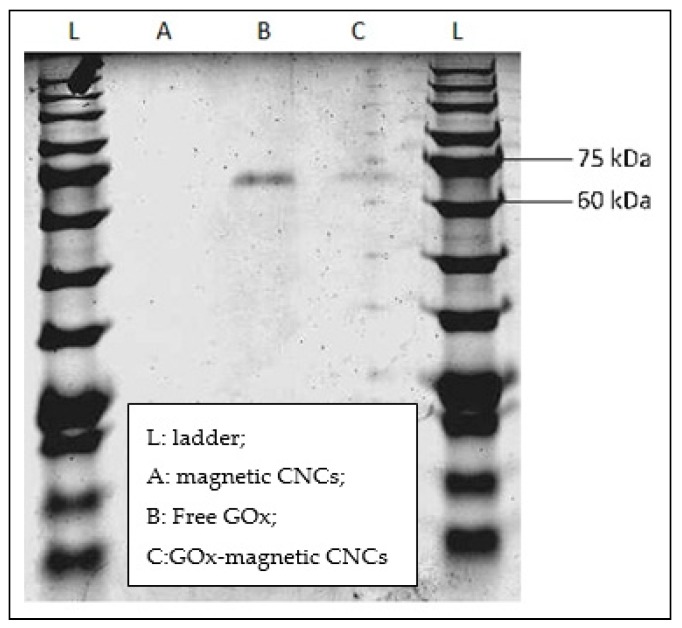
SDS-PAGE showing the attachment of GOx on magnetic CNCs. (**L**: ladder; **A**: magnetic CNCs; **B**: Free GOx; **C**: GOx-magnetic CNCs).

**Figure 4 sensors-19-02511-f004:**
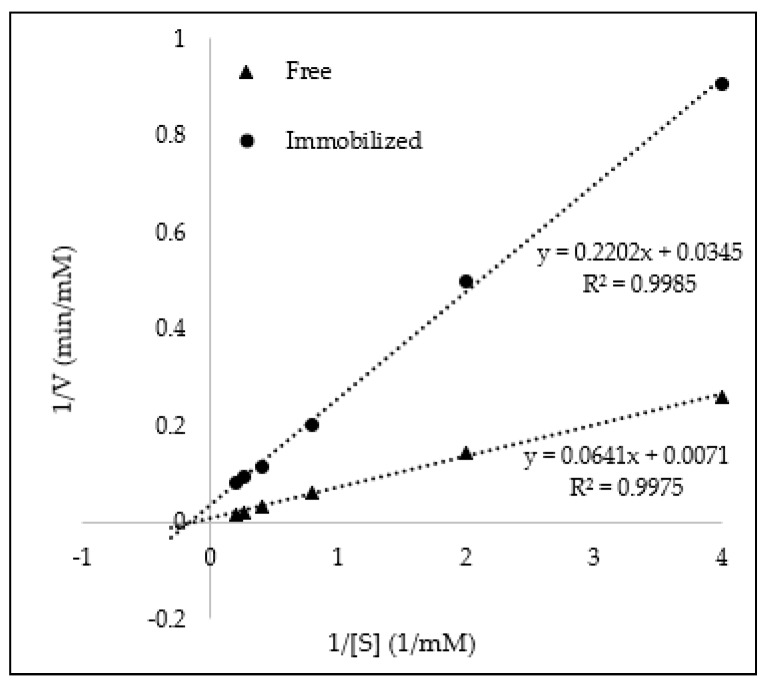
Lineweaver–Burk plot of immobilized GOx activity.

**Figure 5 sensors-19-02511-f005:**
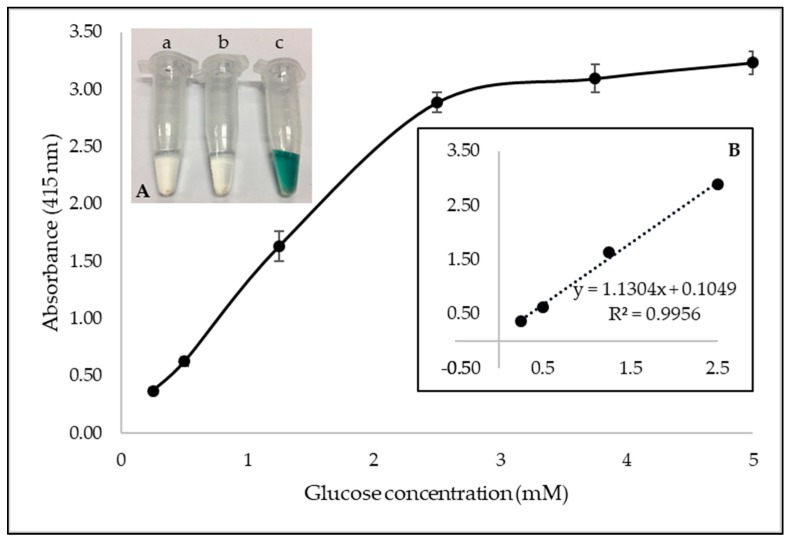
Dose-dependent curve of glucose detection using GOx-magnetic CNCs. (Inset A showing reactions (**a**) without HRP, (**b**), control and (**c**) with 25 mM glucose; Inset B showing the linear range of glucose detection).

**Figure 6 sensors-19-02511-f006:**
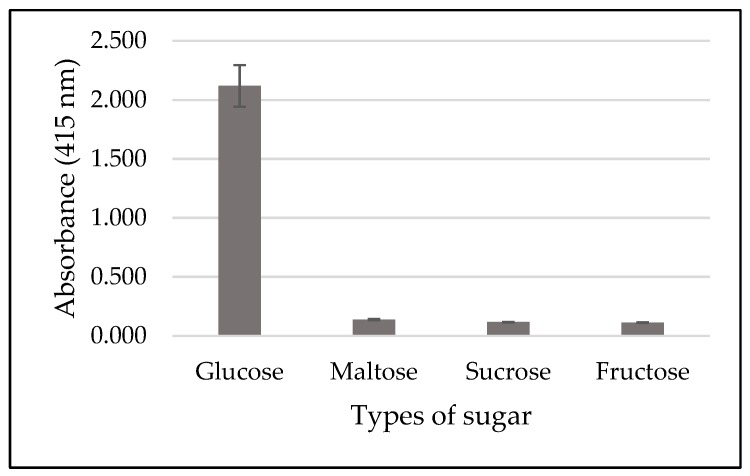
Specificity test of immobilized GOx using glucose (1.25 mM), maltose (1.25 mM), sucrose (1.25 mM), and fructose (1.25 mM).

**Figure 7 sensors-19-02511-f007:**
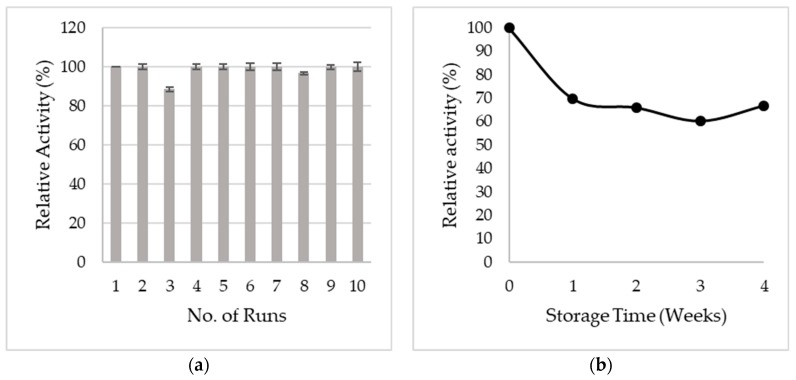
(**a**) Reusability and (**b**) storage stability of GOx-magnetic CNCs. (Error bar represents the standard error of triplicate data).

**Table 1 sensors-19-02511-t001:** Zeta potentials of cellulose nanocrystals (CNCs), iron oxide nanoparticles (IONPs) and magnetic CNCs.

Sample	Zeta Potential (mV)
CNCs	−42.53 ± 1.595 ^1^
IONPs	+28.37 ± 1.002 ^1^
Magnetic CNCs	−22.97 ± 0.306 ^1^

^1^ Standard error was calculated based on triplicate measurements.

**Table 2 sensors-19-02511-t002:** Comparison of different immobilization support on the reusability of immobilized GOx.

Immobilization Support	Reusability	References
Magnetic CNCs	100% of activity retained after 10 cycles.	This study
Fe_3_O_4_ nanoparticles	75% of activity retained after 4 cycles.	[32]
Fe_3_O_4_/SiO_2_ nanoparticles	60% of activity retained after 6 cycles.	[31]
CoFe_2_O_4_/SiO_2_ nanoparticles	57% of activity retained after 7 cycles.	[33]
Fe_3_O_4_ nanoparticles	50% of activity retained after 5 cycles.	[34]

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
