# Peer review of "Colorimetric Analysis of Glucose Oxidase-Magnetic Cellulose Nanocrystals (CNCs) for Glucose Detection"

_sensors, 2019, doi:10.3390/s19112511_

Reviewer 1 Report

 Ying Chuin Yee et al developed a colorimetric glucose detection system using glucose oxidase immobilized on magnetic CNCs. This work is rather typical contribution, not innovative and not well elaborated.

Some of the major criticisms of the manuscript:

  - Quality of figures 1 and 3 are not appropriate.

2.3. Immobilization of glucose oxidase on magnetic CNCs.

What concentrations of EDC and NHS were used?

 - Authors are writing: “After incubation, magnetic CNCs were washed with PBS twice to remove excess reactants, then stored in buffer solution for further use.” Activated carboxylic groups are not stable, so immobilization of GOx have to be performed immediately after activation step.

 - What buffer (concentration, pH) was used for the dissolution of GOx?

Are magnetic CNCs stable if Zeta potential is -22.97 mV?

SDS-PAGE results are difficult to understand, especially line C. These results are not informative for confirmation of GOx immobilization or characterization of magnetic CNC-GOx. Why free GOx and magnetic CNC-GOx lines are at the same place? Have GOx been removed from magnetic CNC-GOx and then SDS electrophoresis was performed? Why many small lines are visible in line C? 

Fig. 5 shows linear dependence of signal at very narrow interval of glucose concentrations (0.25 – 2.5 mmol/l). There are many other possibilities/detection systems to detect glucose in wider glucose concentrations interval. Results at higher glucose concentrations should be presented. How concentration of H2O2 produced during GOx enzymatic reaction was calculated?

Author Response

Hi,

Thank you for your fruitful comment and suggestion, As attached is our response on some of the concern points. 

Reviewer 2 Report

The authors fabricated the GOx-magnetic CNCs for glucose detection. The GOx-magnetic CNCs exhibited the good stability for long-term storage. However, there are a few points that could be included/modified to further improve the quality of the manuscript. I would therefore consider it favorable for publication in Sensors under the provision that the following concerns are adequately addressed.

1. The title is a bit misleading since the color change response was never shown in this colorimetric analysis.

2. The iron oxide nanoparticles showed intrinsic peroxidase-like activity, with catalytic behavior similar to HRP. Thus, these nanoparticles were previously used to catalyze the oxidation of ABTS by H2O2 to the oxidized colored product and provides a colorimetric detection of glucose [1]. In the authors’ work, the ABTS assay was used to investigate the activity of GOx-magnetic CNCs. In Figure 4, I was wondering if the activity might be partially from the iron oxide nanoparticles. In order to eliminate doubts, the authors could test the activity of GOx-magnetic CNCs in the absence of HRP.

Reference

[1]” Intrinsic peroxidase-like activity of ferromagnetic nanoparticles” Nature Nanotechnology 2, 577–583 (2007)

3. The authors stated that “By addition of 60 µg/ml of GOx to magnetic CNCs, only 20.08 µg/ml were successfully immobilized.” How does the authors know that 20.08 µg/ml GOx were immobilized on the magnetic CNCs? Could the authors show the measured results?

4. In Figure 5, the photographs for glucose detection with the colorimetric method developed using GOx and the as-prepared magnetic CNCs could be present. Also, a dose−response curve and detection limit for glucose detection should be given. In addition, the oxidation of ABTS is a colored product. Why does the Y axis not represent the absorption value of ABTS at 415 nm?

5. The concentration information of all components is not given in the figure caption. Please revise it in Figure 6.

6. In Figure 7a, the authors claimed that “GOx-magnetic CNCs maintained 100% of its initial activity even after ten reaction cycles.” Nevertheless, the activity of GOx-magnetic CNCs had a decrease to ~ 90% in Cycle 3. What do you think the reason is? The authors should discuss this part and explain why the activity of GOx-magnetic CNCs comes back to ~100% after 10 cycles.

7. The practical applications in the body fluid should be performed.

Author Response

Hi,

Thank you for the fruitful comment and suggestion. As attached is our response on the concern points. 

Thanks

Reviewer 3 Report

The authors present a colorimetric sensor for glucose based on the glucose oxidase reaction.  The enzyme was covalently bound to magnetic nanoparticles allowing the enzyme to be reusable.  Tests showed the sensor to be reused at least 10 times and fairly stable for a period of 4 weeks.  The authors also showed that the sensor is selective to glucose.

p 1 line 25 - 35 - the introduction seems to downplay the number of studies that use colorimeric detection of glucose using the glucose oxidase chemistry.

p 4 equations 3 and 4 - These equations should include the enzymes that facilitate the reactions.

p 9 section 3.3 - The limit of detection and other analytical metrics should be calculated and included to facilitate comparisons to other methods.

p 10 - Figure 7, what do the error bars represent in this figure?

Why was the peroxidase enzyme not immobilized as well? 

Author Response

Hi,

Thank you for the fruitful comment and suggestion. As attached is our response on the concern points. 

Thanks

Round  2

Reviewer 1 Report

The author has made some modifications. The article can be published in the journal.

Author Response

Hi,

Thank you for the feedback. We highly appreciate it. 

Thanks.

Reviewer 2 Report

I have only one comment in regard to the revised Figure 6. The glucose concentration should be given. I guess the concentration of glucose is between 1.25 and 2.5 mM because the absorbance intensity of ABTS for glucose in Figure 6 falls between the ~1.5 (1.25 mM) and ~3 (2.5 mM).

Author Response

Hi,

Thank you for the feedback. As attached is our response on the raised point.

Thanks.
